# Extreme rainfall events alter the trophic structure in bromeliad tanks across the Neotropics

Gustavo Q. Romero [1✉], Nicholas A. C. Marino [2,3], A. Andrew M. MacDonald[4,20], Régis Céréghino[5], M. Kurtis Trzcinski [6], Dimaris Acosta Mercado[7], Céline Leroy [8,9], Bruno Corbara[10], Vinicius F. Farjalla[3], Ignacio M. Barberis [11], Olivier Dézerald[12], Edd Hammill[13], Trisha B. Atwood[13], Gustavo C. O. Piccoli[14], Fabiola Ospina Bautista [15,16], Jean-François Carrias [10], Juliana S. Leal [2], Guillermo Montero [11], Pablo A. P. Antiqueira [1], Rodrigo Freire [11], Emilio Realpe[15], Sarah L. Amundrud[17], Paula M. de Omena [1,21], Alice B. A. Campos[2], Pavel Kratina [18], Eoin J. O'Gorman [19] & Diane S. Srivastava [17]

Changes in global and regional precipitation regimes are among the most pervasive components of climate change. Intensification of rainfall cycles, ranging from frequent downpours to severe droughts, could cause widespread, but largely unknown, alterations to trophic structure and ecosystem function. We conducted multi-site coordinated experiments to show how variation in the quantity and evenness of rainfall modulates trophic structure in 210 natural freshwater microcosms (tank bromeliads) across Central and South America (18°N to 29°S). The biomass of smaller organisms (detritivores) was higher under more stable hydrological conditions. Conversely, the biomass of predators was highest when rainfall was uneven, resulting in top-heavy biomass pyramids. These results illustrate how extremes of precipitation, resulting in localized droughts or flooding, can erode the base of freshwater food webs, with negative implications for the stability of trophic dynamics.

[1] Laboratory of Multitrophic Interactions and Biodiversity, Department of Animal Biology, Institute of Biology, University of Campinas (UNICAMP), Campinas, SP 13083-862, Brazil. [2] Programa de Pós-Graduação em Ecologia, Universidade Federal do Rio de Janeiro (UFRJ), CP 68020, Rio de Janeiro, RJ, Brazil. [3] Departamento de Ecologia, Instituto de Biologia, Centro de Ciências da Saúde, Universidade Federal do Rio de Janeiro, PO Box 68020, Rio de Janeiro, RJ, Brazil. [4] Centre for the Synthesis and Analysis of Biodiversity (CESAB-FRB), Aix-en-Provence, France. [5] Laboratoire Ecologie Fonctionnelle et Environnement, Université de Toulouse, CNRS, Toulouse, France. [6] Department of Forest and Conservation Sciences, University of British Columbia, Vancouver, BC, Canada. [7] Department of Biology, University of Puerto Rico - Mayagüez Campus, Mayagüez, PR 00681, USA. [8] AMAP, Univ. Montpellier, CIRAD, CNRS, INRAE, IRD, Montpellier, France. [9] UMR ECOFOG, CIRAD, CNRS, INRAE, AgroParisTech, Université de Guyane, Université des Antilles, 97379 Kourou, France. [10] Université Clermont-Auvergne, CNRS, LMGE (Laboratoire Microorganismes: Génome et Environnement), F-63000 Clermont-Ferrand, France. [11] Facultad de Ciencias Agrarias, Instituto de Investigaciones en Ciencias Agrarias de Rosario, IICAR-CONICET-UNR, Universidad Nacional de Rosario, S2125ZAA, Zavalla, Argentina. [12] ESE, Ecology and Ecosystem Health, INRAE, Agrocampus Ouest, 35042 Rennes, France. [13] Department of Watershed Sciences and the Ecology Center, Utah State University, Logan 84322, USA. [14] Department of Zoology and Botany, University of São Paulo State (UNESP/IBILCE). 15054 - 000, São José do Rio Preto, SP, Brazil. [15] Departamento de Ciencias Biológicas, Universidad de los Andes, Bogotá 111711, Colombia. [16] Departamento de Ciencias Biológicas, Universidad de Caldas, Manizales 170004, Colombia. [17] Department of Zoology & Biodiversity Research Centre, University of British Columbia, Vancouver, BC V6T 1Z4, Canada. [18] Queen Mary University of London, School of Biological and Chemical Sciences, London, UK. [19] School of Life Sciences, University of Essex, Colchester, UK. [20] Present address: Université de Montreal, Montreal, Québec, Canada. [21] Present address: Institute of Biological Sciences, Universidade Federal do Pará, Belém, PA, Brazil. ✉email: gqromero@unicamp.br

Climate change is predicted to dramatically alter precipitation regimes and global hydrological cycles[1–3]. Although changes in the spatial distribution of rainfall can both mitigate and amplify differences between dry and wet regions, there is a consensus that many regions will suffer severe impacts of increased variability and magnitude of precipitation[1–4]. These climatic fluctuations can cause extreme hydrological events, such as flooding and drought, which can lead to widespread, though largely unknown, shifts in ecosystem structure and function, particularly in freshwater ecosystems[5–8]. Such events can expose some ecosystems to conditions with no recent historical precedent[6]. Whereas experimental research has focused on incremental changes in mean conditions, fluctuations or extreme events such as floods and droughts may have more profound ecosystem consequences[6]. In addition, most studies simulating precipitation extremes at the multi-site scale have been limited to a single trophic level, particularly producers[5]. Therefore, there is a clear and urgent need to identify which food web compartments (decomposers, primary consumers, predators) are most vulnerable to rainfall fluctuations and extremes[5,6].

Whereas multitrophic research has focused on a single direction of extreme change, especially drought[7–16], the ecological consequences of drought and flooding have rarely been explored in concert[17]. Such studies suggest that drought can substantially alter aquatic[7–15] and terrestrial food webs[16–19], with consequences for community structure and ecosystem function. Drought also weakens trophic cascades and the strength of biological interactions (e.g., competition, predation), and disproportionately threatens top predators, often resulting in communities dominated by smaller organisms[6–13]. However, previous work has provided limited mechanistic understanding of differences in ecosystem sensitivity to global change[5], has been conducted at a local scale[6], and has used dissimilar experimental approaches and methods[5]. This hampers our ability to predict global impacts of drought and flooding on multiple taxa and trophic levels, including standing stock biomass, trophic biomass pyramids[20–22], and biomass fluxes through the food web[23], across large geographic regions.

Standing stock biomass is a common metric in food web research, with trophic structure represented by the distribution of biomass across different trophic levels[21,22]. These biomass pyramids can summarize changes in complexity and biomass flux through food webs[20–22]. Pyramid size and shape exhibit highly variable patterns across different types of ecosystems worldwide[20], but it has been shown that climatic stability can change the shape of biomass pyramids[24]. Thus, quantifying biomass pyramids improves mechanistic understanding of climate change effects on food web structure, resource partitioning, and energy use.

Here we conducted a geographically coordinated experiment[5] in seven sites across Central and South America (18°N to 29°S, Fig. 1a) to investigate the effects of rainfall changes on trophic structure[24,25]. We used natural, detritus-based microcosms (bromeliad phytotelmata) as model systems due to their widespread distribution and ease of manipulation[10–12]. Bromeliad aquatic ecosystems are inhabited by a diverse fauna[10–12], comprising top predators, mesopredators, and detritivores[15,24,25]. We contrasted rainfall-mediated changes in hydrological stability of the study system with the effects of two main rainfall components: (i) the mean daily amount of rainfall, $\mu$; and (ii) distribution of rainfall events around this mean through time, $k$ (i.e., a measure of evenness in the frequency distribution of rainfall; hereafter "frequency"; Fig. 1b). For instance, reductions in both mean daily rainfall (low $\mu$) and the even frequency of rain (low $k$, hereafter "infrequent rainfall") increase the proportion of days that bromeliads are empty of water. Regular (current) variability of the rainfall components in each site were first determined using recent meteorological data (see Methods). We applied a negative binomial distribution to these data to estimate the parameters $\mu$ and $k$. We applied ten levels of $\mu$ (ranging from 0.1 to 3.0) and three levels of $k$ ranging from 0.5 to 2.0 in a fully factorial experimental design at each of our seven sites for a total of $10 \times 3 \times 7 = 210$ food webs in individual bromeliads. This allowed us to compare ambient, baseline conditions ($\mu = 1$, $k = 1$) and extreme fluctuations of rainfall quantity (10–300%) and frequency (50–200%) to average historical levels of daily variability for each site ("Methods").

We hypothesized that precipitation would affect trophic structure in bromeliads by altering hydrology, that is, the temporal dynamics of water within the bromeliad. We therefore also quantified the underlying hydrological dynamics within bromeliads in the field (see "Methods") as a potential proximate driver, and then projected it in multivariate space using principal component analysis, after standardization between sites (Supplementary Table 2, Supplementary Fig. 1). We defined top predators as species without natural predators within the aquatic food web. All bromeliads were open to colonization and extinction throughout the two-month experimental duration and thus communities could dynamically assemble or disassemble.

The multi-site approach allowed us to explore generalities and site contingencies in food web responses to climate change. We expected stronger impacts of rainfall events when their components departed from current scenarios ($\mu \neq 1$, $k \neq 1$). If bigger predators are more sensitive to drought (here measured as lower values of $\mu$ and/or $k$) than smaller organisms (e.g., mesopredators and detritivores)[6–8,24], then drought could have stronger ecological impacts than heavy rainfall[6]. Consequently, under drought or extremely infrequent rainfall events we expected communities dominated by smaller organisms[7] (especially detritivores and filter feeders), which tend to be suppressed via top-down control under more favorable conditions (Fig. 1c). Conversely, an excessive amount of rainfall combined with higher frequency of rainfall among consecutive days could impact lower trophic levels through hydrodynamic perturbations, e.g., if important nutrients and basal resources (detrital organic matter and microorganisms) are lost to flooding[26,27]. Bigger predators could be more resistant to flooding, but their biomass could still decrease if they have fewer resources to support them. Consequently, such changes in standing stock biomass could alter predator–prey biomass ratios (i.e., pyramid shape; Fig. 1d)[20,21,24]. Because drought and high rainfall frequency are predicted to favor lower and higher trophic levels, respectively, drought is expected to be associated with bottom-heavy biomass pyramids, and high rainfall frequency is expected to be associated with top-heavy biomass pyramids (Fig. 1d). Both the climate and the regional species pool are different among sites[12,25,27], which may lead to site-specific contingencies, e.g., communities from arid regions and/or regions with large natural amplitudes in rainfall may be more resistant to drought than those from wet regions and/or regions with narrow climatic amplitudes[27].

Extreme precipitation events and underlying hydrological dynamics influence each trophic level in a different manner. Whereas the biomass of detritivores, often the smaller organisms in the study system, is higher under more stable hydrological conditions, the biomass of predators is highest under uneven rainfall (drought conditions). Higher resource concentration under drought conditions fuels these higher trophic levels, resulting in top-heavy biomass pyramids. Our results demonstrate that organisms from lower trophic levels may be the most susceptible to ongoing climate change.

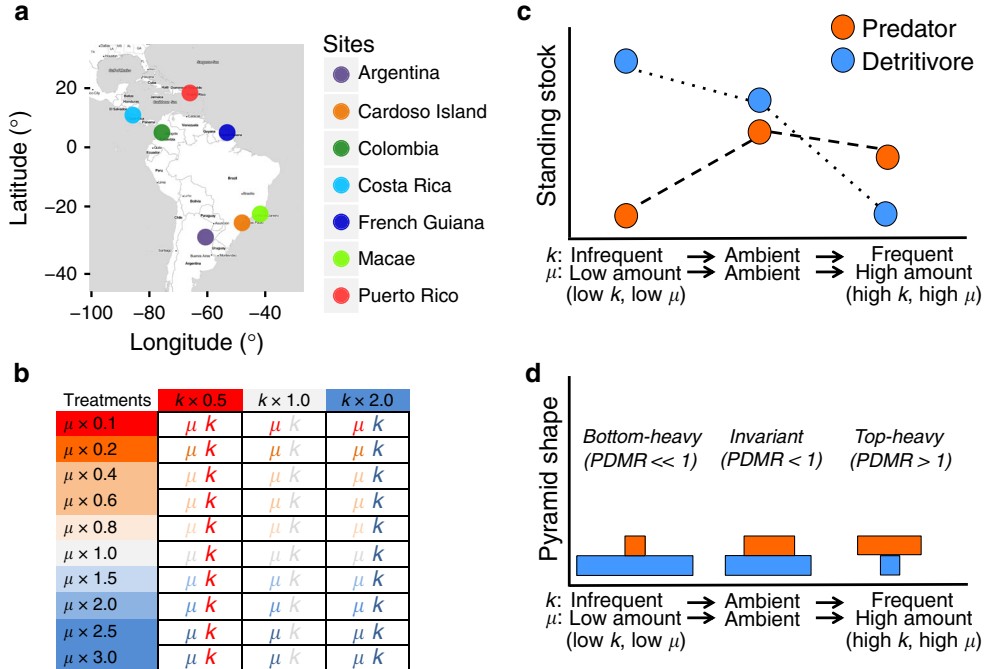

**Fig. 1 Study sites, treatments and conceptual scheme depicting the predictions. a** Study sites. **b** Treatment combinations designed to manipulate the quantity of rain entering the ecosystem (50–200%) and distribution of rainfall frequency (10–300%) relative to average ambient conditions of each site ($\mu = 1$, $k = 1$, in the centre of the table). Gradients of rainfall conditions, from severe drought to frequent rainfall, are represented by red and blue colours. $k$ is the dispersion parameter (a measure of evenness in the frequency distribution of rainfall), and $\mu$ is the mean parameter (a measure of mean daily rainfall). These parameters were calculated per each site based on rainfall patterns of the last five years before the beginning of each experiment (see "Methods"). Stronger impacts of rainfall events are predicted when their components depart from current scenarios. **c** We predict that larger predators are more sensitive in environments experiencing infrequent and low rainfall amount (low $k$, low $\mu$), characterized as extreme drought conditions, than smaller organisms. Under these conditions we expect communities dominated by smaller organisms (detritivores and filter feeders), which tend to decrease in standing stock under more favorable conditions (ambient) via top-down control. In contrast, more frequent rainfall and high rainfall amount (high $k$, high $\mu$), characterized as heavy rainfall, could impact lower trophic levels through hydrodynamic perturbations (e.g., overflow of nutrients and basal resources). Bigger predators could be more resistant to flooding, but their biomass could decrease slightly via bottom-up effects. **d** Changes in standing stock, in turn, could alter predator–prey mass ratios (pyramid shape). Drought, ambient and heavy rainfall conditions could cause bottom-heavy, invariant and top-heavy biomass pyramids, respectively.

## Results

**Standing stock**. Top predators, mesopredators, and detritivores responded differently to rainfall components, and only detritivores were influenced by hydrological stability (Table 1). Predator standing stock biomass decreased with increasing rainfall frequency across all sites in a remarkably consistent response (i.e., no site vs. rainfall interactions, Fig. 2a), despite large differences between sites in average standing stock (Table 1) and taxonomic composition[27]. These emergent patterns indicate that basic properties and processes are recurring in different food webs and deserve deeper understanding. First, top predators account for the majority of predator biomass due to their large body sizes. Second, contrary to initial predictions, total predator biomass (top predators and mesopredators) was not negatively affected by infrequent rainfall. Instead, the standing stock of this trophic group increased as rainfall distribution became more infrequent, and decreased under more even rainfall dispersion (frequent), relative to ambient conditions ($P = 0.045$, Fig. 2a, Table 1). A similar pattern was observed if only top predators were evaluated, though with marginally non-significant results (top predators; $P = 0.067$, Table 1, Supplementary Fig. 2).

In contrast to top predators, detritivore standing stock decreased under extremes of rainfall frequency at some sites and increased in others, thus indicating strong site-specific responses (Table 1, Fig. 2b). In addition, detritivores were more sensitive to hydrological instability, with biomass being higher under more stable hydrological conditions (Fig. 2c). In contrast, mesopredators were not affected by rainfall or hydrological stability (Table 1).

**Biomass pyramids**. Differential responses among the trophic levels resulted in consistent rainfall-driven shifts in the shape of biomass pyramids across sites (i.e., no site vs. rainfall interactions for biomass ratios; Fig. 2d, Table 1). Biomass pyramids comprised of all predators (meso and top predators) consistently became more top-heavy in many communities (i.e., increased total predator-detritivore mass ratio [PDMR]), and even inverted (PDMR > 1), under infrequent (low k) rainfall conditions. A similar pattern emerged for top predator-detritivore ratios (Table 1, Supplementary Fig. 2). This pattern was driven primarily (i) by top predators, rather than by mesopredators (Table 1), and also (ii) by predator body size (Fig. 3) rather than predator abundance. Predator abundance declined with increasing amount of rainfall ($\mu$; Supplementary Fig. 3), but was not influenced by rainfall frequency ($P < 0.05$, backward selection). We also found a strong non-linear increase of PDMR with increase in whole system biomass, but the curves plateaued with accumulating biomass (Fig. 4a). This pattern was determined exclusively by predator biomass (Fig. 4b), whereas an increase in detritivore biomass decreased PDMR (Fig. 4c).

**Table 1 Main statistical results.**

| Predictors | Total predator | | Top predator | | Mesopredator | | Detritivore | |
|---|---|---|---|---|---|---|---|---|
| | $\chi^2$ | P | $\chi^2$ | P | $\chi^2$ | P | $\chi^2$ | P |
| Standing stock (community biomass) | | | | | | | | |
| site | 151.3 | <0.001 | 108.3 | <0.001 | 307.3 | <0.001 | 23.5 | <0.001 |
| k | 6.2 | 0.045 | 5.4 | 0.067 | – | – | 18.5 | <0.001 |
| k:site | – | – | – | – | – | – | 23.2 | 0.01 |
| Hydrology | – | – | – | – | – | – | 6.3 | 0.012 |
| Ecosystem size | – | – | – | – | – | – | 15.7 | <0.001 |
| Model comparisons | | | | | | | | |
| AICc$_{full\ model}$ | 691.9 | | 616.8 | | 216.7 | | 1292 | |
| AICc$_{reduced\ model}$ | 627.9* | | 546.9* | | 126.4* | | 1228.6 | |
| Biomass ratios (predator-detritivore) | | | | | | | | |
| site | 79.5 | <0.001 | 78.4 | <0.001 | 12.1 | 0.059 | | |
| k | 13.1 | 0.001 | 12.3 | 0.002 | 0.02 | 0.99 | | |
| k:site | – | – | – | – | – | – | | |
| Hydrology | – | – | – | – | – | – | | |
| Ecosystem size | – | – | – | – | – | – | | |
| Model comparisons | | | | | | | | |
| AICc$_{full\ model}$ | 359.3 | | 341.9 | | 138.2 | | | |
| AICc$_{reduced\ model}$ | 334 | | 306.5 | | 59.5 | | | |

Generalized linear models (GLMs), following backwards selection, examining the direct and interactive effects of site, ecosystem size (bromeliad volume), rainfall frequency ($k$, 3 levels), quantity ($\mu$, continuous) and hydrological stability (PC1, continuous) on the standing stock of all predators, top predators, mesopredators and detritivores, and on the structure of trophic pyramids (measured as predator-detritivore mass ratios). Linear and non-linear (quadratic) effects of the numerical predictor variables were accounted for in all models. Probabilities were calculated using likelihood ratio tests (LRT, $\chi^2$). Empty cells (—) denote variables removed during backward selection procedure. Full versus reduced (final) models were discriminated using Akaike Information Criteria corrected for small sample sizes (AICc).
*Best fit model included hydrology (PC1), although this variable was not significant in the generalized linear models ($P \geq 0.60$).

## Discussion

We explored the simultaneous impacts of drought and flooding extremes on lentic ecosystems. Our results contrast with findings from lotic ecosystems[8–11], which found higher trophic levels to be more susceptible to drought or hydrological disturbances than lower trophic levels[9,15]. Like many ponds and other wetlands, bromeliads are lentic water bodies that are naturally prone to partial desiccation and overflow. Larger predators from these environments might be better adapted to drought, bouncing back quickly from perturbations. For such lentic ecosystems, flooding events seem to be at least as harmful as drought, as flooding can leach out important nutrients, basal resources (organic matter, microorganisms)[26,27] and even small macroinvertebrates (e.g., detritivores)[27] from the ecosystem. Indeed, we found turbidity, a measure that integrates organic and inorganic suspended matter in ecosystems, including free-living algae and particulate nutrients (C, N, P), resulting from detritivore activity, to be lower under more frequent rainfall conditions in our experiments (Supplementary Fig. 4). This can make ecosystems less productive and thus less efficient at sustaining higher trophic levels. Predators may still be able to persist in such drought conditions because of high biomass turnover by fast-growing organisms[24,25]. Moreover, larger organisms living in intermittent ecosystems may already be selected to withstand drought[12,13]. Since top predators are often the largest aquatic organisms in bromeliad ecosystems, they likely have the highest metabolic demands[28], and thus must maximize foraging in a constrained space. Indeed, some bromeliad-living predators are known to display higher prey capture rates under drought or infrequent rainfall[11,29]. When rainfall is more infrequent ($k < 1$), water levels may drop, concentrating resources (as observed by turbidity results) and benefitting top predators through higher prey encounter rates. This mechanism is not restricted to bromeliad food webs and can be observed in several other intermittent freshwater ecosystems[29,30].

Whereas detritivore standing stock exhibited strong site-specific responses to extremes of rainfall frequency, mesopredators were not affected by rainfall or hydrological stability. The patterns observed for detritivores may be explained by the different ambient rainfall patterns among sites[12,25,27], with indications that communities from some regions are more resistant to drought and flooding than others[26,27]. Conversely, mesopredators are known to be opportunistic species with generalized feeding and habitat requirements and are less sensitive to habitat features and climate change[24,31,32]. Thus, it seems that rainfall extremes modulate different foraging strategies among consumers of different trophic levels. Further experiments should determine if consumption rates and desiccation-related mortality are both negatively related to the rainfall frequency gradient.

Top-heavy biomass pyramids and higher predator:prey body size ratios consistently became more common under infrequent rainfall scenarios. Two major pathways likely generated top-heaviness in our system: (i) increased energy transfer across trophic boundaries within the ecosystem (endogenous pathway) and (ii) increased energy transfer across ecosystem boundaries (exogenous pathway)[21]. Endogenous pathways were evidenced by increasing ecosystem productivity (resource concentration, supplementary Fig. 4) and prey encounter rates[11,29,30] under infrequent rainfall. Likewise, exogenous pathways may promote top-heaviness when biomass turns over more slowly as trophic rank increases[22,28]. Top predators in these ecosystems have long lifecycles (>1 year) relative to their prey (often <1 month for *Culicidae* and *Chironomidae*). Therefore, standing stock of prey may be low, but they normally replenish rapidly in the face of predation pressure through high rates of oviposition by terrestrial adult flies (with such eggs representing terrestrial subsidies[24,25,33,34]). However, infrequent rains reduce the window for oviposition, as insect eggs in this system generally need to be kept humid to survive[10]. This reduction in oviposition quickly

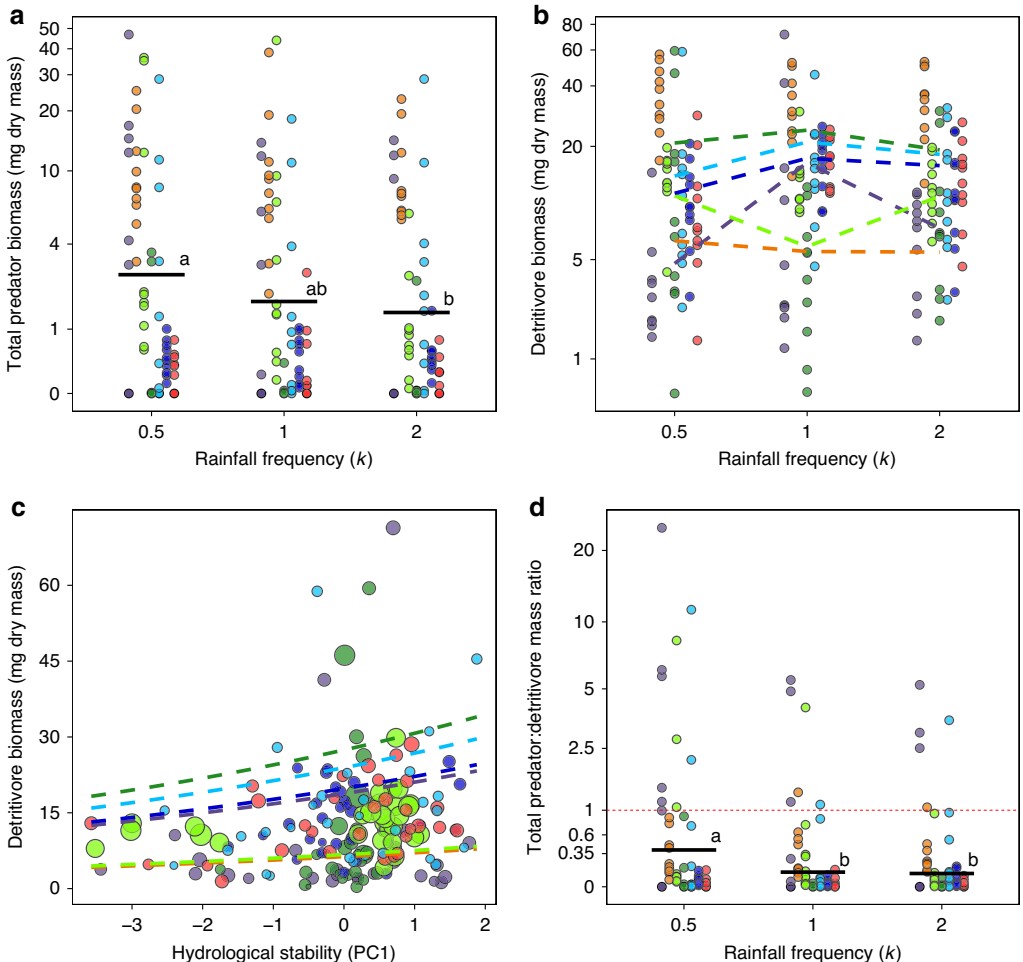

**Fig. 2 Standing stock of predators and detritivores, and pyramid shape, change under rainfall frequency and hydrological stability.** Influence of rainfall frequency (k) on **a** total predator standing stock (top + mesopredators), **b** detritivore standing stock, and **d** total predator:detritivore mass ratio (PDMR; pyramid shape). Detritivore biomass was also influenced by habitat size and hydrology (**c**). Horizontal bars **a**, **d** denote mean values; different letters indicate statistically significant differences (Tukey contrasts, $\alpha = 0.05$). In **c**, habitat size (bromeliad volume) is represented by circles of varying sizes. In **d**, PDMR > 1 (threshold in red dashed line) indicates bromeliads with inverted trophic pyramids. k equals to 1 represents typical average values of rainfall frequency; values lower and higher than 1 represent deviance from typical values, and characterize extreme events. Circle colors indicate sites of the experiment; colors match with the sites plotted in Fig. 1a. Jitter function was used to add random noise to data in order to prevent overplotting.

affects the standing stock of the fast-turnover prey, leading to inverted pyramids, whereas the long-lived predators may survive from an influx of terrestrial prey. Other experiments that reduce oviposition rates have shown similar inversion of biomass pyramids[35]. More generally, as inverted trophic pyramids and higher predator:prey body size ratios are both associated with greater interaction strength and unstable predator–prey dynamics[20,21,36], these findings indicate that extreme reductions in rainfall frequency have the potential to destabilize food webs.

We show how extremes of precipitation affected each trophic level in a different manner. There were remarkably consistent changes in the standing stock of top predators and trophic biomass pyramids in response to altered frequency of rainfall across all sites, despite site-specific differences in detritivore biomass. This implies that similar processes from higher trophic levels may buffer inherent variability within lower trophic levels, and drive consistent food web patterns across the continental scale. In contrast, detritivores were confined to more stable hydrological regimes, greatly affected by rainfall frequency but with strong site-specific contingency. Detritivores either benefited or were impaired by extreme rainfall frequency, whereas large predators mostly benefited from infrequent rainfall. This implies that organisms from lower trophic levels may be more susceptible to rainfall changes in certain geographic regions, and may be restricted to the most stable ecosystems in the near future. As the manipulated changes in extreme rainfall were within predictions for the next few decades[1–4], we can predict strong changes in food web structure and dynamics in the near future[5–8,15,37]. In transient water bodies, such as small streams, pools, ponds or phytotelmata, this may intensify trophic interaction and result in less stable ecosystems[21].

## Methods

**Study sites and experimental communities.** We performed a geographically coordinated experiment[5], manipulating both the amount and temporal distribution of rainfall entering tank bromeliads. Our goal was to investigate the effects of variation in rainfall on macroinvertebrate communities (standing stock at three adjacent trophic levels and the shape of trophic pyramids). We replicated the experiment at seven sites across Central and South America (from 29°S to 18°N), including Las Gamas (Argentina), Cardoso and Macae (Brazil), Colombia, Pitilla (Costa Rica), French Guiana, and Puerto Rico. Detailed site descriptions, the experimental manipulations, and complete list of macroinvertebrate families composing each functional group are provided in ref. [27] and Supplementary Table 3. While the taxonomic compositions of macroinvertebrate communities are

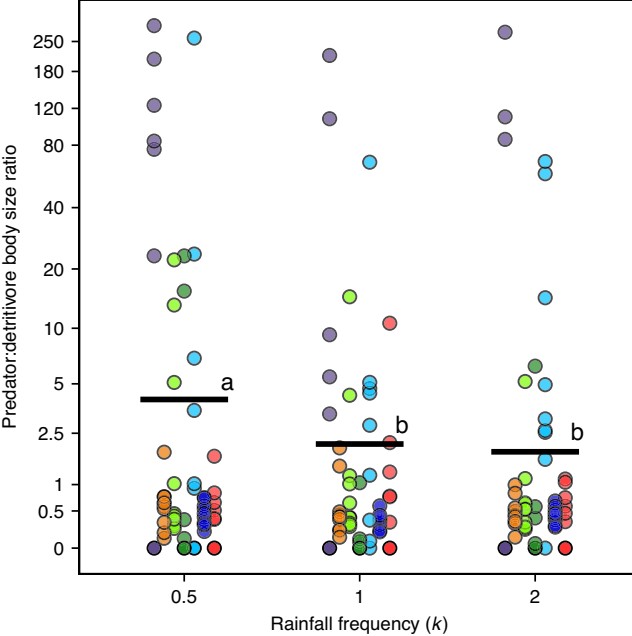

**Fig. 3 Predator:detritivore body size ratio varies across the sites and change under rainfall frequency.** Influence of rainfall frequency ($k$) and site on predator:detritivore body size ratio within each microcosm. Horizontal bars denote mean values; different letters indicate statistically significant differences (GLM/Tukey contrasts, $\alpha = 0.05$). The best model detected site ($\chi^2 = 208.6$, $P < 0.001$) and $k$ ($\chi^2 = 7.1$, $P = 0.028$, $\Delta AICc = 6$) influencing predator abundance, with no significant interaction between these variables. $k$ equals to 1 represents typical average values of rainfall frequency; values lower and higher than 1 represent deviance from typical values, and characterize extreme events. Circle colors indicate sites of the experiment; colors match with the sites plotted in Fig. 1a. Jitter function was used to add random noise to data in order to prevent overplotting.

site specific, all taxa can be assigned to three trophic levels: detritivores, meso-predators, and top-predators[24,25]. The detritivores are typically represented by larvae of Diptera (*Chironomidae* (except *Tanypodinae*), *Culicidae* (except *Tox-orhynchites*), *Syrphidae*, *Tipulidae*), and Coleoptera (*Scirtidae*). The largest top predators are represented by larvae of damselfly (*Coenagrionidae*), horsefly (*Tabanidae*), adult Coleoptera (*Dytiscidae*) and leeches (*Hirudinea*). The meso-predators often include larvae of *Ceratopogonidae* (*Bezzia* spp.), *Corethrellidae*, *Toxorhynchites* (*Culicidae*) and *Tanypodinae* (*Chironomidae*). However, the large predators are not present at all sites[25]. Thus, we defined top predators on a site-specific basis, as the species without aquatic predators themselves. Consequently, mesopredators at some sites can act as top predators in other sites (e.g., *Tox-orhynchites* spp., *Corethrella* spp.).

**Rainfall manipulation**. We established the experimental rainfall treatments for each site using procedures described in ref. [38], Supplementary Note 1 (Supplementary Table 1) and in ref. [27]. Briefly, we used daily rainfall data from the last five years at each site to calculate the site-specific number of days on which a given amount of rainfall was recorded and fit a negative binomial distribution described by the parameters $\mu$ and $k$[38]. The parameter $\mu$ represents the mean daily amount of rainfall and the parameter k represents the frequency of rainfall events around this mean through time (a measure of evenness in the frequency distribution of rainfall). As climate change affects individual sites relative to current conditions, our manipulations of precipitation are intentionally proportional to current site conditions and maintain the temporal autocorrelation structure of each site. The treatment combinations spanned a 30-fold change in $\mu$ and a 4-fold change in $k$ around the ambient conditions, using a response surface design with 30 unique $\mu$ by $k$ combinations (Fig. 1b). The "μ1k1" represents ambient treatment combination (control), while the other treatment combinations were derived by multiplying the control values of $\mu$ by 0.1, 0.2, 0.4, 0.6, 0.8, 1, 1.5, 2, 2.5, and 3, and the control values of $k$ by 0.5, 1 and 2 (Fig. 1b). The range of experimentally imposed values of $\mu$ and $k$ were generally larger than recently observed values in the sites over the experimental months[27]. Thus, extreme values in these ranges represent extreme rainfall conditions. The experiment lasted for 60 days.

**Experimental setup and sampling**. In each site, we selected thirty bromeliads of the most abundant species and with the most common size[27]. We used bromeliads that had more than 100 ml of tank capacity and thus can be colonized by the large predators. We washed each bromeliad with spring water to remove detritus and organisms; we retained coarse (>850 μm) and fine (<850 μm) detritus and sorted macroinvertebrates into species groups. To remove any residual invertebrates, we hung the bromeliads upside down, and let them dry for 7 days. Then, to initiate the community assembly in the experimental ecosystems we evenly divided the fine and coarse detritus between the 30 bromeliads and stocked each bromeliad with the same community in terms of invertebrate families and functional groups[27]. We employed individual transparent plastic shelters above each bromeliad to prevent natural rainfall into the plants. The rain shelters were settled high enough to ensure that they did not alter macroinvertebrate colonization or temperature within the bromeliads. We randomly divided the 30 treatment combinations into three blocks of 10 bromeliads, and initiated each block on one of three consecutive days. This procedure also allowed enough time to sample invertebrates at the end of the experiment.

In order to estimate the key hydrological parameters[15], we measured water depth in the central and two lateral leaf wells of each bromeliad every two days, and used average values per bromeliad. The hydrological parameters for each bromeliad included: (i) coefficient of variation of water depth across the entire experiment, (ii) proportion of overflow days, i.e., the number of days water depth was ≥maximum depth recorded divided by the total number of measurements, (iii) proportion of dried-out days, i.e., the number of days water depth was <5 mm divided by the total number of measurements. These hydrological parameters were used to create a metric of hydrological stability, using Principal Component Analysis (PCA)[15]. We used the scores of the first axis of the PCA (Supplementary Table 2), for each site, to summarize these parameters into a single variable of hydrological stability. This axis quantified a gradient of habitat permanence and stability, where increasing scores represent more stable ecosystems (i.e., ecosystems that dried out less often, and held more water throughout the experiment, Supplementary Table 2).

At the end of the experiment (60th day), we recorded with hand-held data loggers water turbidity, a measure that integrates organic and inorganic suspended matter, including free-living algae and particulate nutrients (C, N, P), resulting from detritivore activity. Thus, turbidity represents a surrogate for total nutrient availability in freshwater ecosystems[39,40]. Then, we dissected each bromeliad by removing and washing each leaf separately in running water and then filtered this water through 125 and 850 μm sieves. We recorded the morphospecies and abundance of all aquatic macroinvertebrates (body size larger than 0.5 mm). We recorded 4–38 morphospecies per bromeliad (mean per bromeliad ± SD: 14.7 ± 9.4). We determined the body size and trophic position of each individual organism surveyed. Trophic position was determined from our own feeding trials, gut contents, stable isotope analyses and from the literature[41–43]. To calculate invertebrate body mass, we used allometric equations between the body length and dry mass, or mean of dry mass for very small insects.

**Statistical analyses**. We tested the independent and interactive effects of site (categorical, seven levels), ecosystem size (maximum volume, continuous), rainfall evenness (k, categorical, three levels) and quantity ($\mu$, continuous) and hydrology (PCA axis 1, continuous), on (i) the standing stock of the three trophic levels, (ii) the structure of trophic pyramids (predator-detritivore biomass ratio), (iii) the predator-detritivore body size ratios, and (iv) ecosystem productivity (turbidity). We used a negative binomial response distribution, which best represents the variation in these responses[27]. Site was treated as a fixed effect in all the models. To account for non-linear effects, we included quadratic functions for all the continuous predictors. Additive and interactive models were analysed using type II and III sum of squares (SS), respectively. Probabilities were calculated using likelihood ratio tests (LRT, $\chi^2$). All the models were reduced using backward selection, and only the final, simplified models are presented. Model simplification consisted of removing more complex non-significant interactions (third order), followed by more simple non-significant interactions (second order), and non-significant quadratic functions[44]. Contingent and consistent responses across all sites were tested as an interaction (linear and quadratic functions) with site, where non-significant interactions indicated consistency across the sites. In these cases, the models were fitted using generalized linear mixed models (GLMM), with site as random factor, for graphical modeling[45].

The data are graphically displayed using the final, reduced (after backward selection) models. All the analyses were performed using R 3.2.2[46]. The established significance level was $\alpha = 0.05$. We checked variance heterogeneity, normality, and outliers through graphic inspections to assure the model assumptions were met. We used the functions *rda* (vegan package) and *prcomp* (stats package) to perform the PCA (hydrological stability). This statistical method differed from a previous analysis of functional feeding groups[27] in that ecosystem size was allowed to leave the model, and backwards selection instead of AIC model selection was employed, so we do not expect results to be completely comparable between the two studies.

**Reporting summary**. Further information on research design is available in the Nature Research Reporting Summary linked to this article.

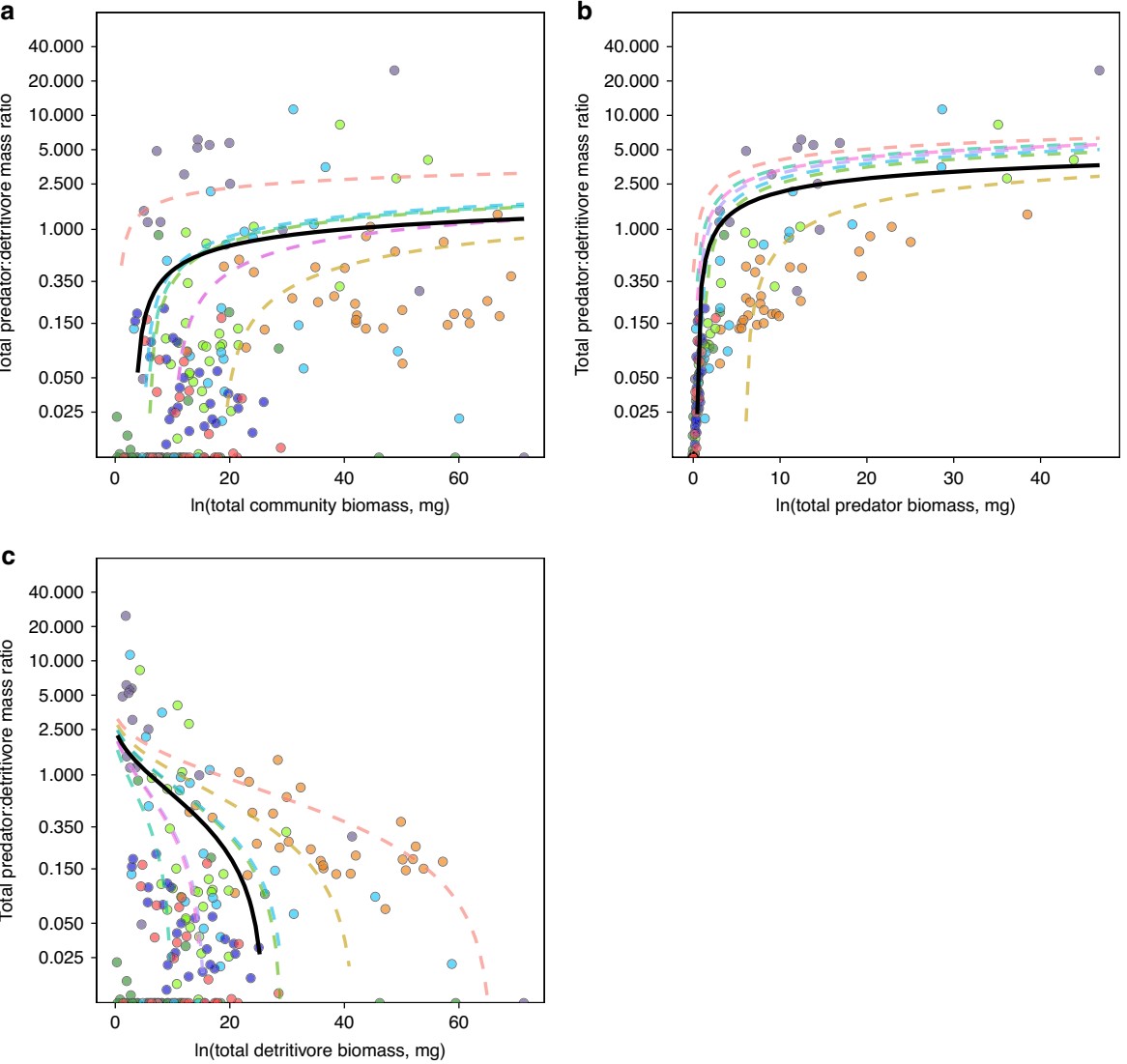

**Fig. 4 Relationships between pyramid shape and biomass.** Relationships between total predator to detritivore mass ratio (pyramid shape) and **a** biomass of the whole community (top predators + mesopredators + detritivores), **b** biomass of total predators (top predators + mesopredators), and **c** biomass of detritivores. All values of biomass are log-transformed (logarithm base e). Circle colors indicate sites of the experiment; colors match with the sites plotted in Fig. 1a.

## Data availability

The data that support the findings of this study are available at https://doi.org/10.5281/zenodo.1124951. This data was collated and hydrologic metrics calculated by a custom-built R package, BWGTools, available at: https://doi.org/10.5281/zenodo.1120418.

## Code availability

The R code used to calculate the precipitation treatments is publically archived at https://doi.org/10.5281/zenodo.18548.

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

## Acknowledgements

We acknowledge financial support for research provided by the São Paulo Research Foundation (FAPESP: grants 2012/51143-3, 2017/09052-4, and BPE Proc. 2016/01209-9) and CNPq through research grants to G.Q.R, by the Agence Nationale de la Recherche (ANR), Rainwebs project (grant ANR-12-BSV7-0022-01) to R.C., C.L., B.C., J.F.C. and D.S.S. and an Investissement d'Avenir grant (Labex CEBA, ref. ANR-10-LABX-25-01), by the Agencia Nacional de Promoción Científica y Tecnológica (grant PICT-2010-1614) and Secretaría de Ciencia y Tecnología de la Universidad Nacional de Rosario (grant AGR-139) through grants to I.B. and G.M., by the Brazilian Council for Research, Development and Innovation (CNPq) for research funds (Pesquisador Visitante Especial, PVE, Research Grant 400454/2014-9) and productivity grants to V.F.F., by the Natural Sciences and Engineering Research Council of Canada (NSERC) through Discovery and Accelerator grant to D.S.S., and the Facultad de Ciencias, Universidad de los Andes, Colombia through a grant (2012-1) to F.O. This research was further supported by scholarship and fellowship support from the ANR to M.K.T., from Consejo Nacional de Investigaciones Científicas y Técnicas (CONICET) to R.F., from FAPESP to P.A.P.A. (Proc. 2014/04603-4, 2017/26243-8), from Coordenação de Aperfeiçoamento de Pessoal de Nível Superior (CAPES) to A.B.A.C., J.S.L., and N.A.C.M. (PNPD-CAPES 2013/0877) and P.M.O. (PNPD-CAPES 2014/04603-4), an Investissement d'Avenir grant (Labex CEBA, ref. ANR-10-LABX-25-01) and a PhD fellowship from the Fond Social Européen to O.D., from NSERC to A.A.M.M. (CGS-D) and D. S.S. (EWR Steacie Memorial Fellowship), from the University of British Columbia to S.L.A., and from the Departamento Administrativo de Ciencia, Tecnología e Innovación (COLCIENCIAS) support of F.O.B. (COLCIENCIAS grant 567). G.Q.R. and P.K. gratefully acknowledge funding from the Royal Society, Newton Advanced Fellowship (grant no. NAF/R2/180791). E.J.O'G. gratefully acknowledge funding from NERC (NE/L011840/1). This paper is a contribution of the Brazilian Network on Global Climate Change Research funded by CNPq (grant #550022/2014-7) and FINEP (grant #01.13.0353.00).

## Author contributions

G.Q.R., D.S.S., R.C., V.F.F., F.O.B., I.M.B., J-F.C., A.A.M.M., B.C., D.A.M., C.L., N.A.C.M., and G.C.O.P. conceived the idea, designed the global experiment, and developed it with M.K.T., O.D., E.H., T.B.A., J.S.L., G.M., P.A.P.A., R.F., S.L.A., P.M.O., and A.B.A.C. D.S.S., A.A.M.M., and N.A.C.M. collated the data and developed hydrological metrics. G.Q.R. drafted the paper with inputs from all authors. G.Q.R. and N.A.C.M. performed the statistical analyses and drafted the figures. G.Q.R., P.K., D.S.S. and E.J.O'G. interpreted the results. A.A.M.M. and D.S.S. created rainfall schedules, G.Q.R., D.S.S., I.M.B., F.O.B., R.C., V.F.F., E.H., D.A.M., G.M., and E.R. organized site-level experiments. All authors participated in discussing and editing the paper.

## Competing interests

The authors declare no competing interests.
