## [Peer Review File · Nature Communications]

Reviewers' Comments:

Reviewer #1:

Remarks to the Author:

This is a very interesting study that adds to previous publications by the same group on the effects of rainfall on aquatic communities in bromeliads. Overall the manuscript is solid and provides information that will advance our understanding of aquatic ecosystems. However, some specific issues need to be addressed:

- The title seems too broad, the study makes a good case for a particular type of freshwater ecosystem, but to stretch to "freshwater ecosystems across the Neotropics" is not appropriate.
- The authors argue that hydrology was not affected by manipulated precipitation. I am not clear at what they mean by hydrology and it is difficult to assess their statement, as it is based on unpublished data. Also, it seems that it might be a point specific for bromeliads. Streams, ponds, and all respond quickly to precipitation. This needs further clarification.
- The authors talk about drought, but as far as I was able to understand from the manuscript, they did not reduce rainfall. How were drought conditions achieved? Systems with low or infrequent precipitation are expected to be adapted to those regimens. Plus, there are different definitions of drought and some do not include seasonal environments where a dry period is expected and organisms are fully adapted to it.
- It would be good to get more details about how they assigned taxa to trophic levels. Identification of aquatic organisms is rather difficult in tropical regions and there is limited information on their biology. Plus, some degree of omnivory is also expected. The methods section lists a very small list of taxa, studies with Neotropical bromeliads often list over 20 taxa.
- Line 246 – there is something wrong here. Allochthonous subsidies and biomass turnover are not related concepts.

Reviewer #2:

Remarks to the Author:

This is an interesting paper with lots of potential. While I like the paper, think it is well written, and find the results interesting the paper to me, as it currently stands, lacks data to make this a more powerful contribution -- as it is, it feels light. Specifically, the results point to the role rainfall frequency may play in mediating trophic structure but it would aid this contribution enormously if there was some more mechanistic data to go with the changing biomass pyramid structure across this gradient. As an example:

- i) are nutrients actually lower in more frequent rainfall scenarios? if so, this natural experiment is showing something very consistent with theory, one trophic level is apparently held in check while the other increases as per Rosenzweig-MacArthur or the paradox of enrichment (a signature of which is increased top heaviness). I know the authors suggest things like this but without some mechanistic details on what is driving their results it seems insufficient for a journal like Nat. Comm.
- ii) are interactions truly stronger (e.g., greater per capita consumption rates) in less frequent rainfall? again their results are consistent with theory on top heaviness. Note, this increased interaction strength in smaller spaces has been also argued to occur in floodplains during their dry season just

prior to migration (check out flood pulse concept)

iii) are both i and ii occurring?

Anything to support one or both of the above or neither would aid this finding.

In summary, I think this paper is moving in a good direction but is light for a significant journal like Nat. Comm. without a more mechanistic underpinning.

General comments and key improvements

We warmly appreciated the referees' suggestions and have followed them closely in our revision. We have now done our best to improve our study following their suggestions.

The main alterations implemented are as follows:

- 1) A change in the title to clarify the type of ecosystem studied.
- 2) Clarifications and corrections of terms used and their consistent use throughout the manuscript.
- 3) Construction of a clear mechanistic framework by providing novel data on ecosystem productivity and predator:prey body size ratio.
- 4) Construction of clearer arguments, with strong support from literature, on prey encounter rates by predators under infrequent rainfall.

Reviewer #1

This is a very interesting study that adds to previous publications by the same group on the effects of rainfall on aquatic communities in bromeliads. Overall the manuscript is solid and provides information that will advance our understanding of aquatic ecosystems.

However, some specific issues need to be addressed:

- The title seems too broad, the study makes a good case for a particular type of freshwater ecosystem, but to stretch to “freshwater ecosystems across the Neotropics” is not appropriate.

Reply: We changed the title to clarify the type of freshwater ecosystem we are studying. The new title is: “Extreme rainfall events alter the trophic structure in natural freshwater microcosms across the Neotropics”.

- The authors argue that hydrology was not affected by manipulated precipitation. I am not clear at what they mean by hydrology and it is difficult to assess their statement, as it is based on unpublished data. Also, it seems that it might be a point specific for bromeliads. Streams, ponds, and all respond quickly to precipitation. This needs further clarification.

Reply: We apologise for the lack of clarity in our statement about hydrology, as we did not mean to imply that rainfall does not affect hydrology. The sentence is not integral to the study, so we have removed it from the revised manuscript. We actually have published evidence that our rainfall treatments affected parameters of hydrology (Srivastava et al. 2020). The reviewer also queried what we meant by hydrology, so we now define it on Line 166 as “the temporal dynamics of water within the bromeliad”.

Srivastava, D., Céréghino, R., Trzcinski, M., MacDonald, A. A., Marino, N. et al. Ecological response to altered rainfall differs across the Neotropics. Ecology e02984 (2020).

- The authors talk about drought, but as far as I was able to understand from the manuscript, they did not reduce rainfall. How were drought conditions achieved? Systems with low or infrequent precipitation are expected to be adapted to those regimens. Plus, there are different definitions of drought and some do not include seasonal environments where a dry period is expected and organisms are fully adapted to it.

Reply: There has been a misunderstanding here: we manipulated both the mean daily amount of rainfall (μ) and the frequency of rainfall events around this mean through time (k). As we explained in Lines 150-160; 557-575 of the manuscript, ambient levels of the rainfall components were first determined using meteorological data from each site. We then applied ten levels of μ (ranging from 0.1 to 3.0) and three levels of k ranging from 0.5 to 2.0 in a fully factorial experimental design. This allowed us to compare ambient conditions ($\mu = 1, k = 1$) to both drought conditions (e.g. 10% quantity, 50% frequency) and extreme rainfall (e.g. 300% quantity and 200% frequency).

To help avoid similar confusion among other readers, we have now moved Table S2, which shows the treatment combinations, from the supporting information to the main text, where it is now Table 1. In addition, we improved this table by including colour gradients denoting extreme drought and frequent rainfall events.

- It would be good to get more details about how they assigned taxa to trophic levels. Identification of aquatic organisms is rather difficult in tropical regions and there is limited information on their biology. Plus, some degree of omnivory is also expected. The methods section lists a very small list of taxa, studies with Neotropical bromeliads often list over 20 taxa.

Reply: We have now included the information about the methods used to assign taxa into trophic levels. The new sentence reads as follows (Lines 541-543): “*Detailed site description, the experimental manipulations, and complete list of macroinvertebrate families composing each functional group are provided in²⁷ and Table S3*”.

We assigned taxa to each trophic level using literature, field observations, and also detailed stable isotope data from most of the sites. This information is presented in Lines 612-615 as follows: “*We determined the body size and trophic position of each individual organism surveyed. Trophic position was determined from our own feeding trials, gut contents, stable isotope analyses and from the literature^{38,39,40}*”.

Regarding the number of taxa (overall richness), we recorded 4 to 38 morphospecies inhabiting individual bromeliads (mean per bromeliad \pm SD: 14.7 ± 9.4). This information has been included in Lines 611-612: “*We recorded 4 to 38 morphospecies per bromeliad (mean per bromeliad \pm SD: 14.7 ± 9.4)*”.

- Line 246 – there is something wrong here. Allochthonous subsidies and biomass turnover are not related concepts.

Reply: We reorganized the information along the paragraphs to make the text clearer. We meant that “*...Predators may still be able to persist in such drought conditions because of high biomass turnover by smaller organisms^{24,25} ... (Lines 252-253)*”, the underlying mechanism is that small prey taxa have very short lifecycles and high colonization rates via egg-laying terrestrial females. Thus, species in these lower trophic levels reach the system via cross ecosystem mechanisms (i.e., oviposition – an allochthonous subsidy from land to water). We have also moved the term “allochthonous subsidies” to another paragraph, where it is now in a better context, and is presented as follows:

Lines 283-294: “*Likewise, exogenous pathways may promote top-heaviness when biomass turns over more slowly as trophic rank increases^{22,28}. Top predators in these ecosystems have long lifecycles (>1 year) relative to their prey (often <1 month for Culicidae and Chironomidae). Therefore, standing stock of prey may be low, but they normally replenish rapidly in the face of predation pressure through high rates of oviposition by terrestrial adult flies (with such eggs representing terrestrial subsidies^{24,25,33,34}). However, infrequent rains reduce the window for oviposition, as insect eggs in this system generally need to be kept humid to survive¹⁰. This reduction in oviposition quickly affects the standing stock of the fast-*

turnover prey, leading to inverted pyramids, whereas the long-lived predators may survive from an influx of terrestrial prey. Other experiments that reduce oviposition rates have shown similar inversion of biomass pyramids³⁵”.

Reviewer #2

This is an interesting paper with lots of potential. While I like the paper, think it is well written, and find the results interesting the paper to me, as it currently stands, lacks data to make this a more powerful contribution -- as it is, it feels light. Specifically, the results point to the role rainfall frequency may play in mediating trophic structure but it would aid this contribution enormously if there was some more mechanistic data to go with the changing biomass pyramid structure across this gradient.

i) are nutrients actually lower in more frequent rainfall scenarios? if so, this natural experiment is showing something very consistent with theory, one trophic level is apparently held in check while the other increases as per Rosenzweig-MacArthur or the paradox of enrichment (a signature of which is increased top heaviness). I know the authors suggest things like this but without some mechanistic details on what is driving their results it seems insufficient for a journal like Nat. Comm.

Reply: The paradox of enrichment and Rosenzweig-MacArthur theory apply to closed systems in which population dynamics occur at the same scale as the scale of observation. By contrast, bromeliads are open systems because the population dynamics of insects, which include adult life stages in the surrounding terrestrial forest, exceeds that of an individual bromeliad. Therefore, such theory cannot be directly applied to our results. However, we do appreciate the point that greater information on basal resources can deepen our study. One of the best measures of available resources is turbidity, which integrates organic and inorganic dissolved matter in bromeliad ecosystems, including some nutrient fractions (particulate N and P), algae and detritus. These new data, presented in the Figure S6, show that turbidity is higher under more infrequent rainfall events. These data are presented in:

Main text; Lines 247-251: “Indeed, we found turbidity, a measure that integrates organic and inorganic suspended matter in ecosystems, including free-living algae and particulate nutrients (C, N, P), resulting from detritivore activity, to be lower under more frequent rainfall conditions in our experiments (Fig. S6).”.

And

Methods; Lines 604-608: “At the end of the experiment (60th day), we recorded with hand-held data loggers water turbidity, a measure that integrates organic and inorganic suspended matter, including free-living algae and particulate nutrients (C, N, P), resulting from detritivore activity. Thus, turbidity represents a surrogate for total nutrient availability in freshwater ecosystems^{39,40}”.

ii) are interactions truly stronger (e.g., greater per capita consumption rates) in less frequent rainfall? again their results are consistent with theory on top heaviness. Note, this increased interaction strength in smaller spaces has been also argued to occur in floodplains during their dry season just prior to migration (check out flood pulse concept)

Reply: We have now included new evidence from one of the experimental sites (Costa Rica), showing that per capita capture rate by bromeliad-living predators are higher under drought and infrequent rainfall.

Lines 255-263: *“Since top predators are often the largest aquatic organisms in bromeliad ecosystems, they likely have the highest metabolic demands²⁸, and thus must maximize foraging in a constrained space. Indeed, some bromeliad-living predators are known to display higher prey capture rates under drought or infrequent rainfall^{11,29}. When rainfall is more infrequent ($k < 1$), water levels may drop, concentrating resources (as observed by turbidity results) and benefitting top predators through higher prey encounter rates. This mechanism is not restricted to bromeliad food webs and can be observed in several other intermittent freshwater ecosystems^{29,30}.”*

We have also included new data on mean predator:prey body size ratio (Figure S3), because these data allow us to partition the response of predator biomass into that driven by body size and abundance and show that body size effects dominate. The mean predator:prey body size ratio also indicate the interaction strength in food webs (see our reference # 36).

(i) Lines 224-232: *“Biomass pyramids comprised of all predators (meso and top predators) consistently became more top-heavy in many communities (i.e., increased total predator-detritivore mass ratio [PDMR]), and even inverted (PDMR >1), under infrequent (low k) rainfall conditions. A similar pattern emerged for top predator-detritivore ratios (Table 2, Fig. S2). This pattern was driven primarily (i) by top predators, rather than by mesopredators (Table 2), and also (ii) by predator body size (Table 2, Fig. S3) rather than predator abundance. Predator abundance declined with increasing amount of rainfall (μ ; Fig. S4), but was not influenced by rainfall frequency (backward selection).”*

(ii) Lines 295-298: *“More generally, as inverted trophic pyramids and higher predator:prey body size ratios are both associated with greater interaction strength and unstable predator-prey dynamics^{20,21,36}, these findings indicate that extreme reductions in rainfall frequency have the potential to destabilize food webs.”*

iii) are both i and ii occurring?

Anything to support one or both of the above or neither would aid this finding.

In summary, I think this paper is moving in a good direction but is light for a significant journal like Nat. Comm. without a more mechanistic underpinning.

Reply: The inclusions and clarifications detailed above allowed us to build a much stronger mechanistic framework, which is presented as a new paragraph (Lines 277-298) of the revised manuscript.

Lines 277-298: *“Top-heavy biomass pyramids and higher predator:prey body size ratios consistently became more common under infrequent rainfall scenarios. Two major pathways likely generated top-heaviness in our system: (i) increased energy transfer across trophic boundaries within the ecosystem (endogenous pathway) and (ii) increased energy transfer across ecosystem boundaries (exogenous pathway)²¹. Endogenous pathways were evidenced by increasing ecosystem productivity (resource concentration, Fig. S6) and prey encounter rates^{11,29,30} under infrequent rainfall. Likewise, exogenous pathways may promote top-heaviness when biomass turns over more slowly as trophic rank increases^{22,28}. Top predators in*

these ecosystems have long lifecycles (>1 year) relative to their prey (often <1 month for Culicidae and Chironomidae). Therefore, standing stock of prey may be low, but they normally replenish rapidly in the face of predation pressure through high rates of oviposition by terrestrial adult flies (with such eggs representing terrestrial subsidies^{24,25,33,34}). However, infrequent rains reduce the window for oviposition, as insect eggs in this system generally need to be kept humid to survive¹⁰. This reduction in oviposition quickly affects the standing stock of the fast-turnover prey, leading to inverted pyramids, whereas the long-lived predators may survive from an influx of terrestrial prey. Other experiments that reduce oviposition rates have shown similar inversion of biomass pyramids³⁵. More generally, as inverted trophic pyramids and higher predator:prey body size ratios are both associated with greater interaction strength and unstable predator-prey dynamics^{20,21,36}, these findings indicate that extreme reductions in rainfall frequency have the potential to destabilize food webs.”.

Reviewers' Comments:

Reviewer #2:

Remarks to the Author:

The authors have responded adequately and I believe this is a solid acceptable piece

Reviewer #3:

Remarks to the Author:

The authors designed an experiment to quantify the effects of extreme precipitation events (droughts and floods) on macroinvertebrate assemblages that reside in bromeliads. The experiment was adequately replicated at multiple sites in the Neotropics, with the aim of having a broader view of fluctuations in precipitation. Standing stock, biomass ratio and body size ratio were used as response variables to rain manipulation.

The results were robust, and the general patterns were maintained at the different Neotropical sites (with some exceptions - detritivore standing stock). During stable hydrological conditions, detritivore biomass was higher, while predator biomass was higher during infrequent rainfall. The authors provide some hypotheses to explain why large numbers of predators were supported under drought conditions.

Looking at the previous revisions, the authors made a great effort to answer, argue, expand, and solidify each of the previous reviewers' comments and suggestions. However, I do not agree with the modification that the authors made to the title. I will argue this below.

Comments to responses to previous reviews.

All comments from previous reviewers were adequately addressed by the authors. I only have one concern with the following comment:

Reviewer #1 suggested modifying the title to avoid being too broad and encompassing all freshwater ecosystems, a suggestion that I agree with. The authors modified the title to include natural freshwater microcosms. However, I wonder whether the word "microcosm" is adequate in this context. When I read the title, I thought the study was a laboratory experiment under fully controlled conditions. I understand that the Authors refer to "natural ...". But, I suggest that the authors continue looking for other more appropriate title options. Here is a potential option: "Extreme rainfall events alter the trophic structure in bromeliad tanks across the Neotropics".

Personal Comments.

Line 96 Are all detritivores small organisms? Are there any large detritivores, either as mesopredators or predators?

I understand that detritivores were classified by feeding trials, gut content, isotopes, and not by size.

Line 125 "... trophic interactions (e.g., competition and predation) ..."

Line 126 As above, I am concerned with referring to all small organisms as detritivores. There are many small organisms that are considered predators, some of which were found in this study (e.g., Ceratopogonidae, Corethrellidae).

Line 134 I am not sure if "currency" is a good adjective for biomass. Why not use "metric"?

Line 144-145 I suggest "...to investigate the effects of rainfall changes on..."

Line 155 This is not very descriptive: ...Ambient levels. Can you use: "Normal variability of..." or "Regular variability of the rainfall components..."

Line 179 "smaller organisms" are not all detritivores.

Line 247 "smaller organisms" are not all detritivores.

Line 253 I suggest: "Predators may still be able to persist in such drought conditions because of high biomass turnover by fast-growing organisms".

Line 270-275 Could the mesopredators be less sensitive to changes in precipitation because their feeding range (i.e., prey) includes microorganisms (not quantified in this study) and macroinvertebrates?

Line 544-547 I do not understand what the authors call detritivores. They have several functional feeding groups in Table S3, but only mention four taxa in this line. Throughout the manuscript, did you only use these 4 taxa for the analyses or did you combine several functional groups (e.g., Filter Feeders, gatherers, shredders) and called them detritivores?

Line 579-580 When the Authors washed the bromeliads, they placed all debris and macroinvertebrates in the same container, and then took out the aliquots? Can you clarify this in the text?

Line 590-592 If the experiment ran for 60 days, during which the authors placed the bromeliads under a shelter, how did they deal with the possible evaporation of water?
Were you constantly replacing water to maintain the treatment (drought and flood) properly?

Reviewer #3

The authors designed an experiment to quantify the effects of extreme precipitation events (droughts and floods) on macroinvertebrate assemblages that reside in bromeliads. The experiment was adequately replicated at multiple sites in the Neotropics, with the aim of having a broader view of fluctuations in precipitation. Standing stock, biomass ratio and body size ratio were used as response variables to rain manipulation.

The results were robust, and the general patterns were maintained at the different Neotropical sites (with some exceptions - detritivore standing stock). During stable hydrological conditions, detritivore biomass was higher, while predator biomass was higher during infrequent rainfall. The authors provide some hypotheses to explain why large numbers of predators were supported under drought conditions.

Looking at the previous revisions, the authors made a great effort to answer, argue, expand, and solidify each of the previous reviewers' comments and suggestions. However, I do not agree with the modification that the authors made to the title. I will argue this below.

Comments to responses to previous reviews.

All comments from previous reviewers were adequately addressed by the authors. I only have one concern with the following comment:

Reviewer #1 suggested modifying the title to avoid being too broad and encompassing all freshwater ecosystems, a suggestion that I agree with. The authors modified the title to include natural freshwater microcosms. However, I wonder whether the word "microcosm" is adequate in this context.

When I read the title, I thought the study was a laboratory experiment under fully controlled conditions. I understand that the Authors refer to "natural ...". But, I suggest that the authors continue looking for other more appropriate title options. Here is a potential option: "Extreme rainfall events alter the trophic structure in bromeliad tanks across the Neotropics".

Reply: we agree, and changed the title as suggested.

Personal Comments.

Line 96 Are all detritivores small organisms? Are there any large detritivores, either as

mesopredators or predators?

I understand that detritivores were classified by feeding trials, gut content, isotopes, and not by size.

Reply: This is correct, we classified the feeding guilds by feeding trials, gut content, isotopes and literature. We did not use body size for these classifications. However, detritivores were almost always smaller than top predators in our system. Overall mean body mass of top predators (1.36 mg) was 7 times higher than of detritivores (0.19 mg). These are average body sizes within each bromeliad, considering all the 210 bromeliads.

Line 125 "... trophic interactions (e.g., competition and predation) ..."

Reply: we changed to "*the strength of biological interactions (e.g., competition, predation)*".

Line 126 As above, I am concerned with referring to all small organisms as detritivores. There are many small organisms that are considered predators, some of which were found in this study (e.g., Ceratopogonidae, Corethrellidae).

Reply: Actually, "smaller organisms" in this sentence also includes small predators (i.e., mesopredators). Please, note that we defined "*top predators as species without natural predators within the aquatic food web*" (lines 166-167).

Line 134 I am not sure if "currency" is a good adjective for biomass. Why not use "metric"?

Reply: we changed to "metric".

Line 144-145 I suggest "...to investigate the effects of rainfall changes on..."

Reply: correction done, thank you.

Line 155 This is not very descriptive: ...Ambient levels. Can you use: "Normal variability of...", or "Regular variability of the rainfall components..."

Reply: We changed to "*Regular (current) variability of the rainfall*"

Line 179 "smaller organisms" are not all detritivores.

Reply: we implemented this sentence as follows: "*If bigger predators are more sensitive to drought (here measured as lower values of μ and/or k) than smaller organisms (e.g., mesopredators and detritivores)^{6-8,24}, then...*".

Line 247 "smaller organisms" are not all detritivores.

Reply: we implemented this sentence as follows: "*...and even small macroinvertebrates (e.g., detritivores)²⁷ from the ecosystem...*".

Line 253 I suggest: "Predators may still be able to persist in such drought conditions because of high biomass turnover by fast-growing organisms".

Reply: correction done, thank you.

Line 270-275 Could the mesopredators be less sensitive to changes in precipitation because their feeding range (i.e., prey) includes microorganisms (not quantified in this study) and macroinvertebrates?

Reply: this is a good idea. We implemented this sentence by including this information. “Conversely, mesopredators are known to be opportunistic species with generalized feeding and habitat requirements and are less sensitive to habitat features and climate change” (lines 275-277).

Line 544-547 I do not understand what the authors call detritivores. They have several functional feeding groups in Table S3, but only mention four taxa in this line. Throughout the manuscript, did you only use these 4 taxa for the analyses or did you combine several functional groups (e.g., Filter Feeders, gatherers, shredders) and called them detritivores?

Reply: indeed, we used these categories (detritivores, mesopredators, top predators) to classify general trophic levels. But within each trophic level we can find several feeding guilds. In the supplementary Table 3, we classified the feeding guilds, and “engulfers” and “piercers” denote predators. The others are detritivores.

Line 579-580 When the Authors washed the bromeliads, they placed all debris and macroinvertebrates in the same container, and then took out the aliquots? Can you clarify this in the text?

Reply: We rephrased some sentences of this paragraph for clarifications. It is as follows: “We washed each bromeliad with spring water to remove detritus and organisms; we retained coarse (>850 μm) and fine (<850 μm) detritus and sorted macroinvertebrates into species groups. To remove any residual invertebrates, we hung the bromeliads upside down, and let them dry for seven days. Then, to initiate the community assembly in the experimental ecosystems we evenly divided the fine and coarse detritus between the 30 bromeliads and stocked each bromeliad with the same community in terms of invertebrate families and functional groups”.

Line 590-592 If the experiment ran for 60 days, during which the authors placed the bromeliads under a shelter, how did they deal with the possible evaporation of water?

Were you constantly replacing water to maintain the treatment (drought and flood) properly?

Reply: no, the bromeliads were underwent to the treatments (please, see the fig 1b), including extreme drought conditions. The roofs (rain shelters) helped to keep the bromeliads under each treatment condition. But no bromeliad tank was completely dried during the experiment due to evaporation.